# External acidity as performance descriptor in polyolefin cracking using zeolite-based materials

Sebastian Rejman [1], Zoé M. Reverdy [1,2], Zeynep Bör[1], Jaap N. Louwen [1], Carolin Rieg [1], Joren M. Dorresteijn [1], Jan-Kees van der Waal [3], Eelco T. C. Vogt [1], Ina Vollmer [1] ✉ & Bert M. Weckhuysen [1] ✉

Thermal pyrolysis is gaining industrial adoption to convert large volumes of plastic waste into hydrocarbon feedstock. However, it suffers from a high reaction temperature and relatively low selectivity. Utilizing a catalyst in the process, moving from thermal pyrolysis to catalytic cracking could help overcome both challenges. In order to develop efficient catalyst materials for this process, understanding structure-composition-performance relationships is critical. In this work, we show that in contrast to cracking of small molecules, plastic cracking activity using ultrastable zeolite Y materials does not depend on the bulk Brønsted acid site content, but rather on the concentration of acid sites located on the outer surface and in mesopores. This external acidity, however, fails to capture all the observed performance trends. Detailed kinetic experiments reveal that the scaling of the reaction rate with the catalyst loading differs drastically between highly similar catalyst materials. More specifically, doubling the catalyst loading leads to doubling of the reaction rate for one material, while for another it leads to more than fivefold increase. When very bulky reactants, such as polyolefins, are converted over micro-porous catalysts, structure-composition-performance relationships established for smaller molecules need to be revisited.

Plastics represent more than 40% of the annual output of the global chemical industry[1]. In order to convert large amounts of plastic waste into useful products, new processes with the required scale need to be developed. A process that meets this criterion and is gaining industrial application is pyrolysis. Its major downside, however, is the high required temperature (>500 °C) and poor selectivity[2]. In crude oil refining, both problems were overcome by the move from destructive distillation to catalytic cracking[3]. In a similar fashion, cracking catalysis could enable the processing of polyolefin plastic waste at scale with improved selectivity. The first commercial plant for catalytic cracking of plastics with a 50 kiloton per year capacity was recently announced[4]. In order to develop improved catalysts for the catalytic cracking of

polyolefins, the properties affecting their performance must be understood. To take the first step towards this goal, we considered the structure-composition-performance relationships established over the past 50 years in the cracking of small hydrocarbon molecules. For the workhorse of cracking catalysis, namely zeolites like zeolite Y (ZY), these relationships have been studied extensively. Cracking catalyst performance is most commonly assessed by determining the first order rate constant for cracking of a small hydrocarbon, like hexane[5-10], or by micro-activity testing (MAT), in which vacuum gas oil (VGO) is converted[11]. The two metrics have been correlated[11]. The activity of a zeolite in the cracking of smaller hydrocarbons depends on a plethora of parameters: The concentration of acid sites[5,6], their

[1]Inorganic Chemistry and Catalysis, Institute for Sustainable and Circular Chemistry, Department of Chemistry, Utrecht University, Utrecht, The Netherlands. [2]ENS de Lyon, Département de Chimie, Lyon, France. [3]TNO, Delft, The Netherlands. ✉e-mail: i.vollmer@uu.nl; b.m.weckhuysen@uu.nl

strength[7], confinement, and adsorption effects[12,13], as well as mass transport[8]. In addition, depending on the zeolite and the reaction conditions chosen, mono or bimolecular cracking mechanisms can dominate[9,10]. The level of understanding for the cracking of polymers, such as polypropylene (PP), is, however, limited. The chemistry of the reaction is thought to be the same as with smaller alkanes[14]: The complex network of reactions is initiated by the protonation of an alkane by a Brønsted acid site (BAS), yielding carbenium ions which can react in multiple ways. Isomerization can lead to a more stable carbenium ion. Carbon-carbon bonds are subsequently broken via beta scission, yielding shorter olefins and alkanes. The carbenium ion can also abstract a hydride from a second alkane, leading to propagation between molecules[15]. The question remains however, how moving to considerably larger hydrocarbons affects the relative prevalence of the different reactions possible in hydrocarbon cracking.

A major obstacle in studying the plastic cracking reaction is the difficulty in obtaining reliable kinetic information. Since plastics are solid at room temperature and highly viscous when molten, most studies rely on thermogravimetric analysis (TGA) approaches, which, in general, can only yield apparent kinetic parameters. For example, for the thermal pyrolysis of polyethylene (PE), apparent activation energies ($E_a$) between 200 and 300 kJ/mol and compensation effects were reported[16]. A higher $E_a$ is offset by a higher preexponential factor. If the logarithm of the preexponential factor ($A$) is plotted against $E_a$ a linear relationship with a slope of $1/RT^*$ is observed, where $R$ is the ideal gas constant and $T^*$ is the mean of measurement temperatures. This type of compensation behavior is common in heterogeneous catalysis[17,18] and was also observed in the cracking of multiple hydrocarbons over a variety of zeolite-based materials[19]. Caution is warranted when interpreting this type of result, as apparent compensation relationships can be caused by both mathematical and experimental errors, and there is no general agreement on how compensation effects are to be interpreted[17,18]. In the case of cracking of lighter hydrocarbons, a compensation effect was explained by kinetics being dominated by the adsorption behavior, for which a linear relationship between activation enthalpy and entropy was shown[19].

Returning to catalytic cracking of polymers, such as PE and PP, prior work showed that for zeolite ZSM-5[20,21] and for mesoporous zeolite Beta[22] the activity in PE cracking increases with increasing external or mesopore surface area. Different zeolite topologies were also compared with each other[14], however it remains a challenge to disentangle the effects of pore structure from the effects of acidity. If a clear link between activity in small molecule and polyolefin cracking could be established, it would indicate that structure-composition-performance relationships determined for the former still hold for the latter, thus allowing the use of literature findings to predict the rates for polyolefin cracking.

In the first part of this work, we tested the above hypothesis by comparing the activity in the cracking of 2,4-dimethylpentane (DMP) and PP over ZY samples of varying acidity. PP of low molecular weight ($M_w$ = 27,000 g/mol) was utilized, as for realistic, high $M_w$ polymers the viscosity of the melt was shown to be the limiting factor[23]. To be able to study the effect of the catalyst itself, this study, therefore, assumes that this problem is overcome by a viscosity reducing pre-treatment in analogy to visbreaking[24] in crude oil refining. For plastic cracking, we initially relied on ramped TGA. While this analytical method does not yield intrinsic kinetic parameters, it still allows us to compare the relative activity between the different catalyst materials under study. Our experiments reveal that activity in small molecule cracking does not predict activity in the cracking of PP, demonstrating that new structure-composition-performance relationships need to be established, which was our aim in the second part of this study. For this, we probed the concentration of BAS located outside of the zeolite micropores by infrared spectroscopy (IR) of tri-tert-butyl pyridine (TTBP). The external acidity established in this way acts as a

significantly better descriptor for plastic cracking activity than bulk acidity. However, the external acidity fails to capture all plastic cracking activity trends, which necessitated an in-depth investigation of the reaction kinetics. Relying on isothermal measurements rather than the more conventional ramped TGA experiments, we show that for seemingly similar zeolite materials, catalyst loading affected the cracking rate very differently. As extensive catalyst characterization failed to explain this observation, we turned to a statistical simulation of the PP cracking process to investigate the effect of the location of cracked chemical bonds along the backbone on the cracking kinetics. Lastly, we studied the effect of the catalyst acidity on the selectivity of the reaction using semi-batch reactor experiments.

## Results and discussion
### Assessment of activity
A set of 5 zeolite-Y samples of varying $SiO_2/Al_2O_3$ (SAR) ratios, further denoted as $ZY_5$-$ZY_{55}$, were obtained from Zeolyst (CVB400-CBV780). For CBV760, two separate batches were utilized ($ZY_{56}$ and $ZY_{47}$), and an additional catalyst was prepared by mild steaming and acid leaching of $ZY_{14}$ (denoted $ZY_{35st}$), leading to a total set of 7 catalyst materials. All zeolites with the exemption of $ZY_5$ exhibited mesoporosity as a result of a steam treatment performed by the supplier to obtain the various SAR. Supplementary Tables S2 and S3 show the results of all catalyst characterization on these catalyst materials under study.

To assess the activity in small hydrocarbon cracking, 2,4-dimethylpentane (DMP) was converted in a packed bed reactor, while polypropylene (PP) cracking was probed using ramped thermogravimetric analysis (TGA) at varying catalyst loading following an approach described previously[23].

Figure 1a shows the first-order rate constant of DMP cracking in units of catalyst mass as a function of bulk BAS content probed by infrared spectroscopy of adsorbed pyridine (Py-IR). For steamed zeolite materials, that contain both strong acid sites and mesoporosity, the rate constant $k_{DMP}$ increased with increasing BAS content. The unsteamed ZY shows significantly lower activity, despite having the highest BAS content. The core reason for the significant increase in the activity of ZY upon steaming has been the subject of debate[25]. However, improved mass transport due to mesoporosity[26,27], and increased acid site strength due to isolation of acid sites and interaction with extra-framework aluminum[28] stand out as the most significant contributors. A higher acid site strength for steamed zeolites was also observed in this study by temperature-programmed desorption of pyridine (Supplementary Fig. S1). Furthermore, a recent work[25] found that silanol groups present for steamed zeolite-Y can shuttle previously inaccessible protons from BAS located inside sodalite cages, noticeably increasing the share of acid sites able to partake in the reaction. The same catalysts show a completely different activity trend in plastic cracking. Figure 1b shows the temperature of the highest cracking rate $T_{max}$ at different catalyst loading obtained by ramped TGA heating at 5 °C/min. Full weight loss profiles can be found in Supplementary Fig. S2. We demonstrated the reproducibility of this approach in a prior study[23], and show it here for zeolites as well (Supplementary Fig. S3). All steamed catalysts exhibited remarkably similar $T_{max}$. However, $ZY_{55}$, which shows the lowest activity in DMP cracking, exhibited the lowest $T_{max}$ and, therefore, the highest activity in the set of as-received zeolites, while $ZY_{14}$, the most active catalyst in DMP cracking, shows the lowest activity of the steamed catalyst in PP cracking. The only consistent trend between the two performance measurements is that steamed zeolite materials are, in general, more active, which was confirmed by testing another non-steamed ZY (Supplementary Fig. S3).

This result implies that activity trends obtained by cracking of shorter model hydrocarbons like hexane or DMP have limited predictive power for polyolefins cracking and is evidence of deviating structure-composition-performance relationships between the two

chemical reactions, which will be elucidated in this work. We recently demonstrated that plastic cracking is subject to multiple types of transport limitations[23], and we hypothesized that polymers enter the micropores of ZY very inefficiently or not at all. This hypothesis is supported by multiple key findings in the literature. Most critically, Macko and coworkers determined the capacity of a CBV780 Zeolite (ZY$_{55}$ in this study) in the adsorption of PP from solution in units of gram of polymer per gram of zeolite[29]. Combining their capacity with the pore volume of our sample and the bulk density of polypropylene, we estimate that only around 5% of the total pore volume or 10% of the micropore volume would be occupied by the polymer. We note that adsorption from the solution will most likely be easier than from the melt, as the polymer chains in the solution are not as entangled. Therefore, the polymer most likely only 'plugs' micropore entries, without properly entering the pore system. Furthermore, side chains resulting from branching might also sterically prohibit the polymer form entering beyond a certain length, in analogy to pore mouth catalysis[30]. Drawing on evidence from gas phase cracking, the diffusion coefficient of linear hydrocarbons in a silicalite molecular sieve decreases exponentially with increasing carbon number[31], suggesting that diffusion will be severely limited for polymers. Furthermore, it was demonstrated that the viscosities of polymers can increase up to two orders of magnitude under confinement[32]. PP cracking activity might, therefore, be more dependent on the number of acid sites located outside of micropores, that is in mesopores and the external surface of zeolite crystals. To test this hypothesis, we probed the external acidity of the zeolite catalysts using 2,4,6-tri-tert-butyl pyridine (TTBP) as a probe molecule, which has previously been shown not to enter ZY micropores[33,34]. In a similar fashion, di-tert-butyl pyridine has been utilized to correlate the cracking activity of bulky hydrocarbons to external acidity for small pore zeolites[35]. Fig. 1c shows difference spectra of TTBP adsorbed from the gas phase, while Fig. 1d shows the quantified bulk and external Brønsted acidity probed by Py-IR and TTBP-IR spectroscopy, respectively. The spectra show the appearance of the N-H band of protonated TTBP, as well as a decrease in the silanol and O-H vibrations of the zeolite material. The decrease in the silanol (3741 cm$^{-1}$) vibration and low-frequency O-H vibration (3560 cm$^{-1}$, assigned to BAS located inside sodalite cages) is caused by a proton transfer mechanism demonstrated recently for steamed zeolites interacting with strong bases[25]. While the bulk acidity consistently decreases with increasing SAR, the external acidity is remarkably similar for all untreated catalyst materials. This explains the similarity in the observed PP cracking activities (Fig. 1b) and demonstrates that external acidity is a more appropriate descriptor for polyolefins

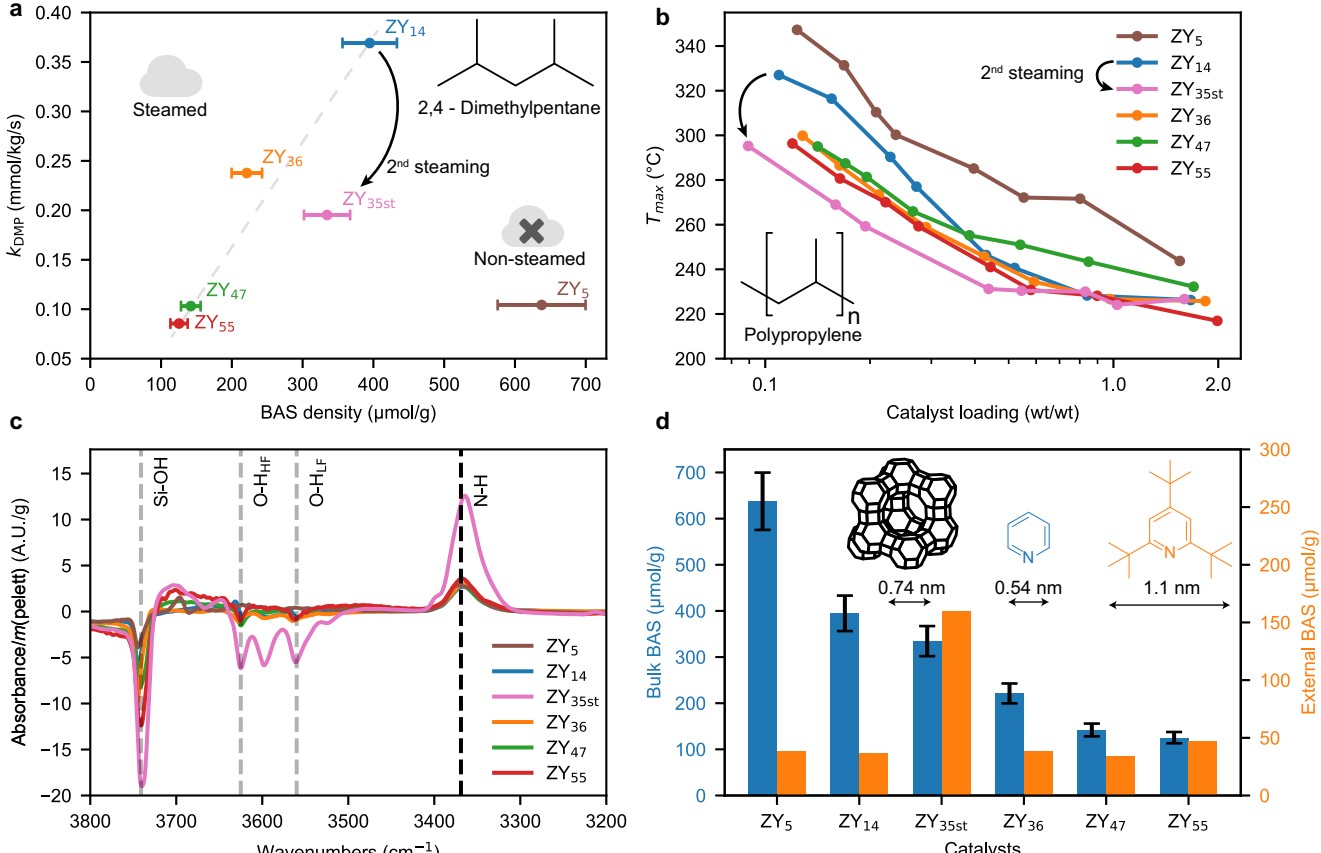

**Fig. 1 | Comparison of small molecule (2,4-dimethylpentane, DMP) and plastic (polypropylene, PP) cracking activity and external acid site characterization.** **a** First order rate constant of DMP cracking as a function of Brønsted acid site content determined by pyridine infrared (IR) spectroscopy for 5 steamed and 1 non steamed zeolite Y. The index in the zeolite label indicates the SiO$_2$/Al$_2$O$_3$ ratio. See Supplementary Fig. S1 for the IR spectra of adsorbed pyridine. Error bars taken from the error of the extinction coefficient[55]. **b** Temperature of highest cracking rate $T_{max}$ in cracking of PP determined by ramped thermogravimetric analysis (TGA) of the polymer-catalyst mixtures at different catalyst loadings. A lower $T_{max}$ indicates higher cracking activity. See Supplementary Fig. S2 for full TGA profiles.

ZY$_{56}$ was omitted for clarity, for all catalysts see Fig. S3. **c** IR spectra of tri-tert-butylpyridine (TTBP) adsorbed from the gas phase on ZY$_x$, showing changes in zeolite O-H vibrations and vibrations of the probe molecule. High-frequency (HF) and low-frequency (LF) indicate high and low-frequency O-H vibrations of acid sites. The spectra of the dried zeolites were subtracted. **d** Bulk and external Brønsted acidity of ZY$_x$ probed by IR spectroscopy of adsorbed pyridine (ring stretch vibration mode at 1544 cm$^{-1}$) and TTBP (N-H stretch vibration mode at 3369 cm$^{-1}$), respectively with the pore diameter of ZY and the kinetic diameters of both probe molecules. No error for the extinction coefficient of TTBP was reported[33].

conversion activity, although not without restrictions, as e.g., $ZY_{14}$ and $ZY_{55}$ show similar external acidities, but different activities at low loading (vide infra). To show how this result can be used to synthesize more active cracking catalysts, we aimed at introducing more external acid sites on a given catalyst. $ZY_{14}$ was subjected to mild steaming (500 °C, 2 h, followed by $HNO_3$ leaching) yielding the $ZY_{35st}$ sample. The steaming treatment reduces the DMP cracking activity by reducing the bulk acidity (Fig. 1a). However, this treatment also results in a more than fourfold increase in external acidity (Fig. 1d), which led to a reduction in $T_{max}$ by more than 40 °C at low catalyst loading (Fig. 1b). Apparently, a treatment that reduces gas cracking activity can noticeably increase plastic cracking activity. This shows that external acidity can serve as a design criterion in the development of new polyolefin cracking catalysts.

The importance of external acid sites in the process sparks the question to what extent their chemistry deviates from acid sites located inside micropores. DFT calculations showed that for the cracking of hexane on low-aluminum ZY, external acid sites exhibit an activation barrier that was only 25 kJ/mol higher than over acid sites inside the zeolite micropores. For comparison, the barrier in the zeolite micropores of high-aluminum ZY was 60 kJ/mol higher than over external BAS on the same zeolite (see Supplementary Note 1 for details). This shows that external acid sites have comparable strength to 'internal' sites, which was also recently shown for zeolite MFI[36]. For zeolite ZY, the strength of individual acid sites is determined primarily by the SAR, rather than the location of the acid site.

Importantly, the external acidity fails to capture the remaining activity trends. $ZY_{55}$ and $ZY_{14}$ showed similar external acidities, yet their $T_{max}$ at low catalyst loading deviated by > 20 °C (Fig. 1b). Similarly, the non-steamed $ZY_5$ shows comparable external acidity, but noticeably lower activity compared with the other zeolites. However, here the decreased activity might be explained in a similar fashion as its reduced gas cracking activity (Fig. 1a), namely by the lower amount of mesoporosity of the non-steamed $ZY_5$ compared to the other zeolites and the higher acid strength of steamed zeolites. Furthermore, the relative PP cracking activities depend on the catalyst loading. The $T_{max}$ curves for $ZY_{14}$ and $ZY_{47}$ intersect at a catalyst loading of 0.3 wt/wt. At low catalyst loading, $ZY_{47}$ is more active, while at higher loading $ZY_{14}$ showed a lower $T_{max}$. To study the remaining differences in activities, we opted to conduct more in-depth kinetic experiments.

## Isothermal cracking of polypropylene

Ramped TGA experimentation is a very convenient and common technique in the study of polymer degradation, and when multiple catalyst loadings are investigated many qualitative insights can be drawn from a few experiments. However, modeling of these kinetics is challenging, requiring a large number of parameters[23,37]. We, therefore, performed additional isothermal TGA to obtain more easily interpretable kinetic data. Instead of ramping the temperature, catalyst-polymer mixtures were quickly heated to a desired cracking temperature, and the weight loss was tracked over time. This experiment was conducted at least 30 times for each catalyst varying the temperature (230 °C– 250 °C) and catalyst loading ([C] = m(catalyst)/m(PP) = 0.1-0.5).

Figure 2a shows the logarithm of isothermal weight loss profiles for $ZY_{14}$. Above ~ 25% conversion, the logarithm of the weight loss profile decreased linearly, indicating the reaction can be modeled using first-order kinetics. The deviations from the first order and the potential mechanistic implications are discussed in Supplementary Note 2. From the slope, the apparent first-order rate constant $k'$ can be determined. To account for the increase in rate with increasing catalyst loading, $k'$ was decomposed into catalyst loading independent rate constant $k$ and a component for the catalyst loading according to Eq. 1. $n$ describes a pseudo-order in catalyst loading. Assuming the first order in catalyst loading did not allow us to describe the experimentally

observed trends. Furthermore, different slopes of the $T_{max}$ curves seen in Fig. 1b show that changes in catalyst loading affect the rate differently for each catalyst.

$$k' = k \cdot [C]^n \qquad (1)$$

Substituting Arrhenius' equation and taking the logarithm yields Eq. 2:

$$\ln(k') = -\frac{E_a}{R} \cdot \frac{1}{T} + n \cdot \ln([C]) + \ln(A) \qquad (2)$$

Where $R$ is the ideal gas constant, $E_a$ is the activation energy, and $A$ the pre-exponential factor. By determining $k'$ at different temperatures and catalyst loadings, a fit of a plane yields all 3 kinetic parameters. Figure 2b shows planes fitted to kinetic data. For an interactive 3D view the reader is referred to the companion Jupyter notebook. A good fit of the plane to each individual dataset demonstrates that the model captures the kinetic features adequately. The isothermal measurements reproduce the trends obtained by ramped TGA. Figure 2c shows $k'$ calculated at 250 °C using the fitted kinetic parameters that show the same activity trends as seen in Fig. 1b: $ZY_5$ showed the lowest rate constant, while at low loading $ZY_{55}$ was the most active catalyst. The 'crossing point' for $ZY_{14}$ and $ZY_{47}$ was reproduced, showing the reliability of the observed differences in activity. The isothermal experiments rely on at least 30 isothermal including different temperatures and catalyst loadings rather than 7 ramped experiments corresponding to different catalyst loadings for each catalyst. $E_a$ increased slightly with increasing Al content from $80 \pm 8$ kJ/mol for $ZY_{55}$ to $101 \pm 13$ kJ/mol for $ZY_5$. However, the confidence ellipses overlap significantly (Fig. 2d), indicating the reactions could be described by very similar kinetic parameters. With $80 \pm 8$ to $101 \pm 13$ kJ/mol, the $E_a$ determined for plastic cracking was significantly lower than for DMP cracking with $140 \pm 6$ to $180 \pm 2$ kJ/mol (Fig. 2e), showing that temperature has a lower effect on the rate of the plastic cracking reaction. While the simplified kinetic model only yields an apparent activation energy, it might still be a sign of a different mechanism. At conditions of low pressure and high temperature, DMP cracking occurs mostly via monomolecular cracking, while plastic cracking was tested at lower temperatures and inherently higher hydrocarbon concentrations.

Therefore the polyolefins cracking reaction could be mainly propagated by hydride transfer, which has been shown to express a lower energy of activation compared to monomolecular cracking[15]. Due to the shear size of plastic molecules, one initiation event can lead to a high number of cracking events, as the carbenium ion is always in proximity to a large supply of hydrocarbons. Furthermore, we studied a pair of catalyst materials for which we previously were able to show clear differences in mass transport, namely a pristine and a crushed fluid catalytic cracking (FCC) catalyst. Crushing the catalyst alleviates accessibility issues caused by the thick outer shell of the FCC particles[23]. For the pristine FCC catalyst, which shows a higher degree of mass transport limitations, an $E_a$ of $33 \pm 21$ kJ/mol was determined, which is 85 kJ/mol lower compared to the crushed catalyst (Supplementary Fig. S1). Such a noticeably lower apparent $E_a$ is a sign of mass transport limitations[38]. The viscosity of PP scales with temperature following an Arrhenius-like shift-factor equation with an apparent activation energy of 40 kJ/mol[39]. This provides kinetic evidence for a hypothesis from earlier work which stated that for a pristine FCC catalyst, the plastic cracking reaction is limited by capillary intrusion[23], which in turn is limited by the viscosity according to Washburn's equation[40]. From this, we propose that if, in a plastic cracking reaction an $E_a$ close to the shift factor activation energy of the polymer viscosity is determined, the reaction is limited by capillary intrusion. If this is

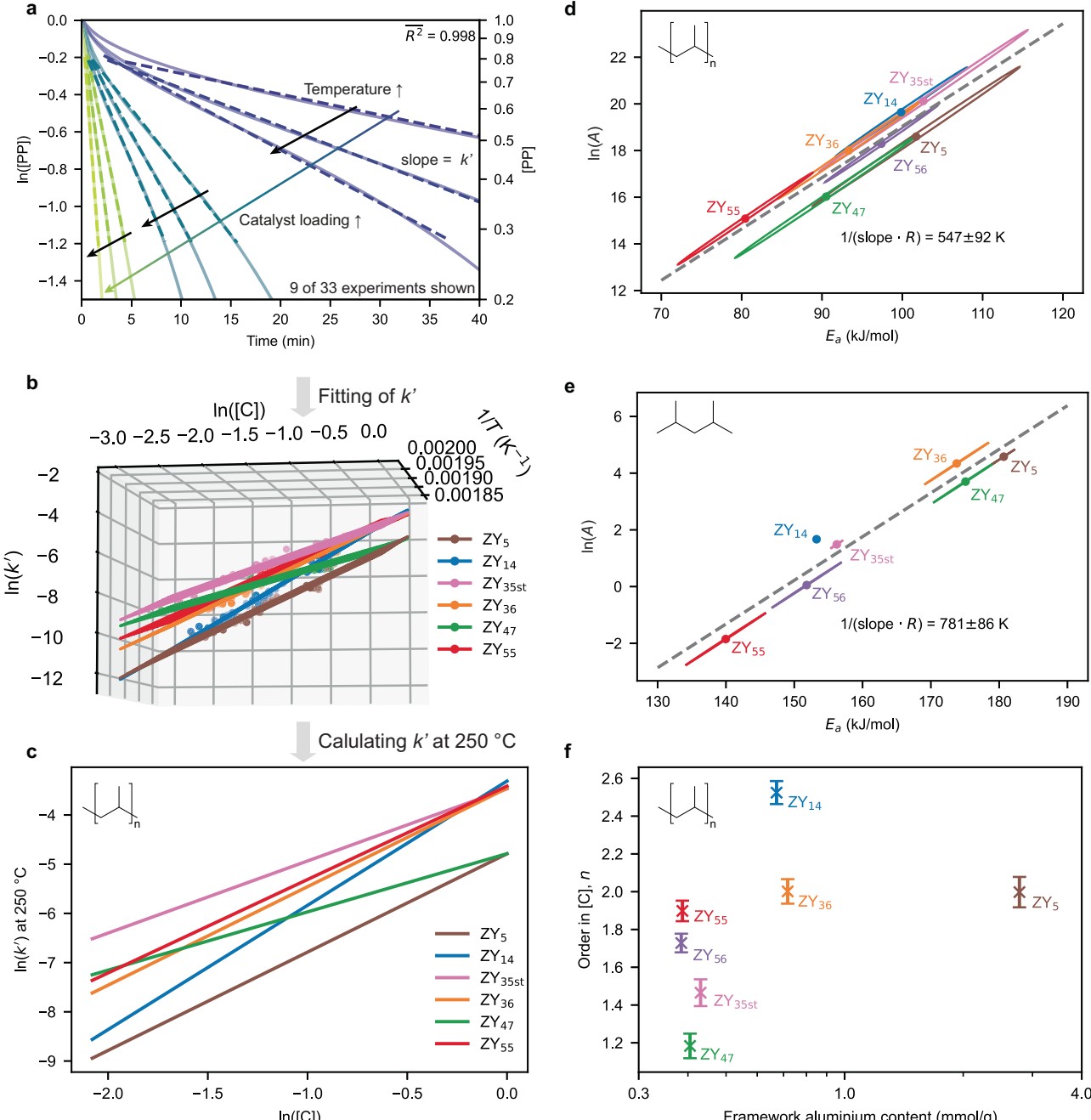

**Fig. 2 | Isothermal thermogravimetric kinetics for cracking of polypropylene (PP) using zeolite Y. a** Logarithm of normalized PP weight over time of cracking experiment using the $ZY_{14}$ zeolite material at different catalyst loadings (0.1–0.5) and temperatures (230–250 °C). The apparent first-order rate constant $k'$ was determined from the slope of a linear fit in a conversion regime of 25–75% (dashed line). See Supplementary Fig. S4 for all catalyst materials under study. **b** $\ln(k')$ as a function of inverse temperature and logarithm of the catalyst loading. A plane was fitted to each dataset according to Eq. 2. The slopes yield the activation energy $E_a$ and the pseudo-order in catalyst loading $n$. Data for the $ZY_{56}$ zeolite material are omitted for clarity. **c** Calculated $k'$ from fitting results according to Eq. 2 at 250 °C as

a function of catalyst loading. **d**, **e** Constable plots showing the compensation relationship between $E_a$ and $\ln(A)$ for PP and DMP cracking, respectively using $ZY_x$. See Supplementary Fig. S5 for Arrhenius plots of dimethylpentane (DMP) cracking. Linear fit using orthogonal distance regression was used to determine the characteristic temperature $T^*$ from the inverse slope. Confidence ellipses are drawn around one standard deviation determined from the covariance matrix of the fits. **f** $n$ for all catalysts studied plotted as a function of framework aluminum content determined from inductively coupled plasma optical emission spectroscopy (ICP-OES) and $^{27}$Al magic angle spinning (MAS) nuclear magnetic resonance (NMR). Error bars show one standard deviation determined from the covariance matrix of the fit.

applied to the zeolites under study, activation energies of > 80 kJ/mol indicate that the reaction is not limited by capillary intrusion. This could imply that polymer macromolecules are not entering the micropores, potentially not even the mesopores of the catalyst.

A compensation effect between $E_a$ and $\ln(A)$ was observed for both reactions. For polyolefins cracking, fits of Eq. 2 yield correlation factors > 0.98 between the two parameters. This can be seen more clearly in a plot of $\ln(A)$ as a function of $E_a$ (Fig. 2d), called a Constable

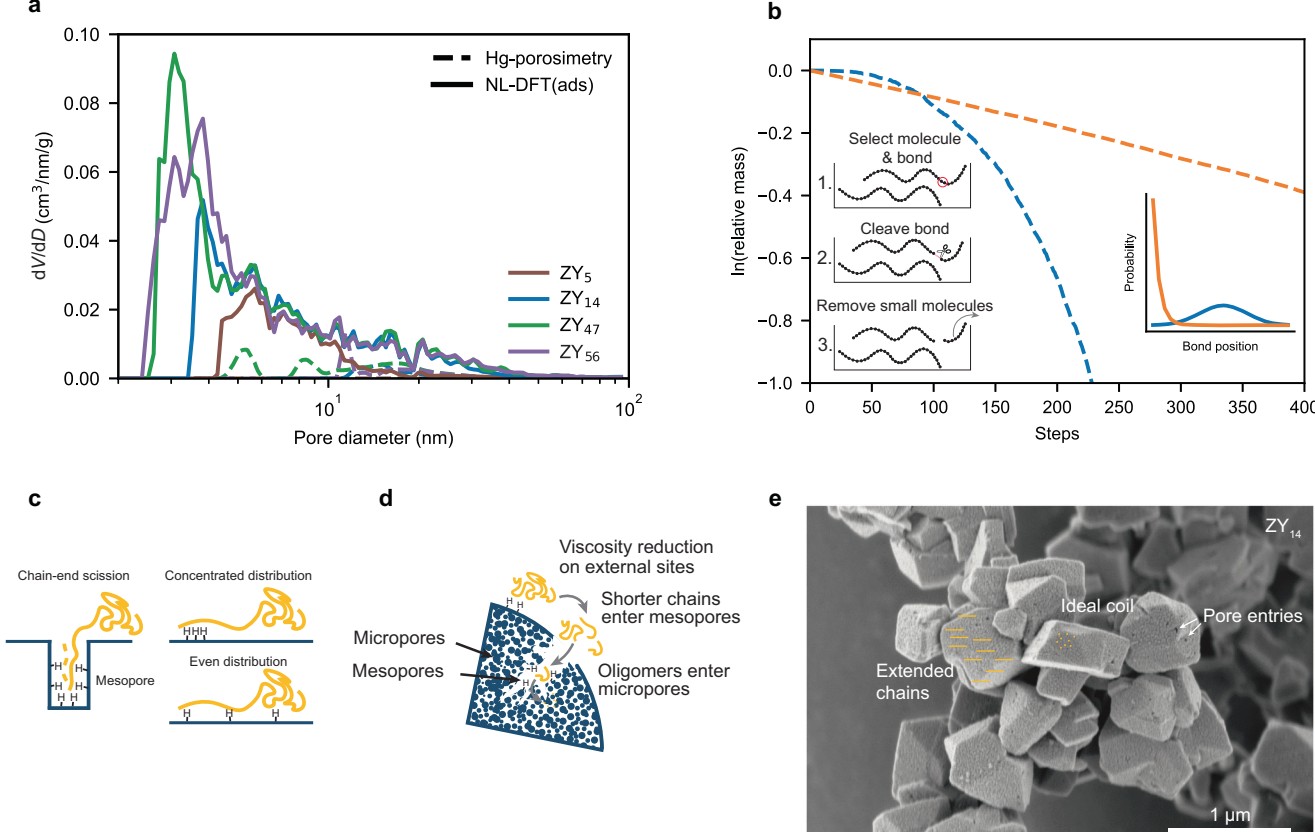

**Fig. 3 | Rationalizing differences in pseudo-order in catalyst loading. a** Pore size distribution for 4 selected zeolites determined by non-local density functional theory (NL-DFT) of the adsorption branch of Ar physisorption at 87 K using a hybrid kernel of spherical micropores and cylindrical mesopores as well as Hg porosimetry. See Supplementary Fig. S6 for all catalyst materials under study, which were omitted for clarity. **b** Simulated weight-loss profiles for a plastic cracking reaction. The bonds to be cracked were selected randomly using different probability distributions: In blue, the probability follows a Gaussian distribution around the middle of the chain, in orange the probability decreases exponentially from the chain end. First-order kinetics in bonds broken were enforced. Insert shows schematically the steps of the simulation. **c** Cartoons depicting how different locations of acid sites could determine which bonds along the polymer backbone are cracked. The polymer chain is yellow, and the catalyst surface in blue. **d** Cartoon showing how acid sites in different locations could work in tandem by consecutive viscosity reduction leading to a self-accelerating effect. **e** Scanning electron micrograph of $ZY_{14}$ particles. Extended chains and ideal coil sizes for polypropylene (PP) with an $M_w = 23,000$ g/mol are drawn schematically to illustrate relative length scales. See Supplementary Fig. S7 for the scanning electron microscopy (SEM) images of all catalyst materials under study.

plot. The characteristic temperature of $T^* = 547 \pm 80$ K ($274 \pm 80$ °C) determined from the slope of the Constable plot ($1/RT^*$) is within the margin of error of the mean of the measurement temperatures (242 °C). In contrast to the compensation relationship seen for DMP cracking (Fig. 2e), the strongly overlapping confidence ellipses indicate that for plastic cracking the compensation is an artifact[17,18]. For DMP the resulting compensation effect can be explained by adsorption effects as described previously for hexane cracking[19]. We, therefore, do not interpret differences in $E_a$ and $\ln(A)$ further.

Instead, we will focus on the pseudo-order in catalyst loading $n$, which shows drastic differences between the catalyst materials (Fig. 2f). For $ZY_{47}$, an order of $1.19 \pm 0.06$ was determined, while for $ZY_{14}$, $n$ was $2.53 \pm 0.06$. This means that for the former, doubling the catalyst loading leads to approximately a doubling in cracking rate, while for the latter catalyst, it leads to a more than fivefold increase in rate. The differences in the scaling of the activity with catalyst loading also present themselves in the ramped TGA experiments. For catalysts with a higher $n$, the $T_{max}$ curve increases more steeply with a decrease in catalyst loading (Fig. 1b). No trend of $n$ with regards to the zeolite aluminum content or any acidity metric either internal or external could be established, and $n$ was uncorrelated to $\ln(A)$ or $E_a$ (correlation coefficients of 0.01 and $-0.03$, respectively.).

## Rationalizing the varying effect of catalyst loading

To rationalize these very large differences between the catalysts, we first aimed at excluding the effects of accessibility as the sole cause. The mesopore structure of the catalysts was studied by non-local density functional theory (NL-DFT) from Ar physisorption at 87 K using the adsorption branch and a hybrid kernel of spherical micropores and cylindrical mesopores. See Supplementary Fig. S6 for the isotherms. Hg porosimetry was employed as an additional technique. Figure 3a shows the results for 4 selected catalyst materials for pore diameters between 2 and 100 nm (see Supplementary Fig. S6 for the remaining catalysts, which are not shown for clarity). The mesopore volume determined by Hg porosimetry was significantly lower than determined by physisorption, which could be explained by the constriction of the mesopores (Supplementary Note 3)[26]. The pore size distributions were very similar for all steamed catalysts, with the exemption of $ZY_{14}$, which shows less mesopores between 2-3 nm diameter, likely because this catalyst was only steamed once, not twice, as is the case with $ZY_{36-55}$[41]. The unsteamed $ZY_5$ showed even less mesoporosity determined by NL-DFT and almost no detectable Hg intrusion in this pore size regime, in line with barely observed hysteresis in its Ar isotherm (see Supplementary Fig. S5). There is no clear trend between the pseudo-order $n$ and the pore size distribution, especially for $ZY_{47}$ and

$ZY_{56}$, which exhibit different $n$ at very similar pore size distributions. In addition, for the crushed and pristine FCC catalysts, both materials show comparable $n$ (Supplementary Fig. S16). From this, we conclude that mass transport effects are not the principal factor determining the scaling of observed activity with changes in catalyst loading. We were not able to find a consistent correlation between $n$ and a variety of catalyst properties, including particle size determined by scanning electron microscopy (SEM) (Supplementary Fig. S7) which was very similar for all catalysts, and combinations of porosity and acidity metrics, e.g., the volumetric and surface density of acid sites in the mesopores.

The observed weight loss kinetics are not purely determined by the rate at which chemical bonds are cleaved. The location at which the polymer is cleaved along the backbone is also critical[42]. This can be shown by a simplified simulation of the process: We assumed that at the start a 'reactor' contains 10 polymer chains of 200 identical carbon units. In each timestep, a chain was selected with increasing probability depending on its length, followed by the selection of a bond to be cleaved according to probability distributions representing two limiting cases: In one limiting case, a hypothetical catalyst material enabled mostly chain-end scission, which was modeled by the probability of selection decreasing exponentially from one chain end. In the second limiting case, a second hypothetical catalyst enabled mostly middle-of-chain scission, which was modeled by a Gaussian around the middle of the chain (Fig. 3b, insert). After the selection of the bond, it was cleaved with a probability depending linearly on the number of bonds in the reactor, enforcing first-order kinetics. All short molecules below a specified length were then removed from the reactor, the remaining 'mass' was determined and the next timestep initiated. The simulation ran 5 times, and the resulting weight loss profiles averaged. Details can be found in the SI, while the Python code can be found in the companion Jupyter Notebook. Figure 3b shows the results of the simulation for the two limiting cases. At the very beginning, the cracking rate for the 'chain-end' model was higher, which was expected since small molecules are formed and leave the 'reactor' right from the start. However, the 'middle-of-chain' model quickly began to react faster, as with few scissions a large number of molecules just below the cutoff length were formed. Therefore, if a catalyst cleaves different bonds along the backbone, it can lead to a difference in weight loss kinetics even if the turn-over frequency (TOF) of the active sites per bond remains the same. This further complicates kinetic analysis of the polyolefins cracking process, as from pure weight loss data it cannot be discerned if an increase in rate is due to higher TOF of the active sites or related to a change in cracking location. We, therefore, are unable to conclusively explain the origin of variations in $n$. However, we suspect that the exact location and distribution of (external) acid sites might play a critical role. If the reaction proceeds largely in mesopores, a processive mechanism[43] with predominantly chain end scission could be dominating (Fig. 3c). On the outermost surface, extended polymer chains could stretch over a large part of the zeolite particle. Figure 3e shows a scanning electron micrograph of $ZY_{14}$ particles, overlayed bars, and dots showcasing the approximate length of an extended or coiled-up polymer chain to illustrate the relative sizes of both.

We expect, therefore, that a single polymer macromolecule can interact with multiple acid sites simultaneously. This becomes more evident if the average distance between external acid sites is estimated (Supplementary Note 4) and compared to an ideal coil or extended chain length of the polymers: For $ZY_{14}$, the average distance between external sites was estimated at 5 nm. By comparison, an ideal coil of PP with an $M_w$ of 27,000 g/mol shows a diameter of 13 nm, while an extended chain is 163 nm long[23]. Therefore, a different arrangement of acid sites could have an influence on the location of cracked bonds along the polymer backbone. If active sites are accumulated together, more chain end scission could be expected, while a more

homogeneous distribution could cause more evenly distributed cracking probability along the polymer backbone (Fig. 3c). Potential testing of this hypothesis by selectivity analysis is discussed in the following section.

Lastly, pre-cracking at the outermost surface of the catalyst material could lower the molecular weight and the viscosity of the melt enough for product molecules to react further in the micropores of the catalyst. Therefore, an increase in external acidity in combination with a high concentration of acid sites in micropores could lead to self-reinforcing effect, which would result in pseudo-orders > 1. The degree of this self-reinforcement will be a result of an interplay between external and bulk acidity (Fig. 3d).

## Selectivity analysis

Analysis of the reaction selectivity can be a powerful tool in shedding light on the reaction mechanism. Furthermore, the acidity of the zeolite material might be a tunable parameter that could steer the selectivity of the reaction. To analyze the effect of acidity on reaction selectivity, PP cracking was conducted in a semi-batch reactor on a 2.5 g PP scale using a catalyst loading of 25 wt%, similar to a previously described procedure[23,44]. The temperature was raised at 10 °C/min to 450 °C under $N_2$ flow. Liquid products were caught in cold traps containing icewater and analyzed using two-dimensional gas chromatography with flame ionization and mass spectrometry detectors (GCxGC-FID-MS), while non-condensed products were quantified by on-line GC-MS-FID. The coke yield was determined by the TGA of the spent catalyst materials. The reproducibility of the approach is shown in Supplementary Fig. S8. Figure 4a shows the evolution of non-condensed hydrocarbons over the reaction time for 4 studied catalysts. For profiles of the individual hydrocarbons, we refer to Supplementary Fig. S9. The evolution profiles of non-condensed hydrocarbons are very similar for all catalyst materials under study, further demonstrating the highly similar activities for the set of catalysts. The slight increase in light hydrocarbon yield with decreasing SAR can be explained by the increased gas-cracking activity which cracks condensable hydrocarbons into shorter molecules. Figure 4b shows the overall yields of the reaction. Deviations from 100% were caused by minor condensation in cold sections of the autoclave which are therefore not collected in the cold traps. The selectivity is again very similar for the catalyst studied with the exemption of coking. Coke formation increases with increasing BAS content (Fig. 4c), which was also observed in ramped TGA cracking (Supplementary Fig. S10 and Supplementary Note 5). This indicates that coke formation occurs largely as a secondary reaction from primary cracking products which can enter into micropores. Aromatic formation, on the other hand, appears to not be affected significantly by the SAR, suggesting aromatics are formed early in the process, and not purely as a secondary reaction, in agreement with a previous study that utilized in-situ spectroscopy[45]. This implies that tuning the reaction selectivity towards products like aromatics cannot be achieved by simply adjusting the bulk acidity of ZY in a semi-batch reactor. The size distribution of the resulting products might provide evidence for preferential cleavage of different bonds along the polymer backbone as discussed in the previous section. For more chain-end scission, one would expect overall lighter products. However, the formed product vapors can still react with the surrounding zeolite, including its micropores, and would therefore be cracked further. In this case, the product size distribution would be noticeably affected by the gas-cracking ability of the given catalyst. As mentioned, more light gases are observed for the catalyst with the highest gas cracking activity, and when qualitatively comparing boiling point distributions of the alkane and alkene products in the pyrolysis oil by masked and integrated 2D-chromatograms (Supplementary Fig. S13), it can also be seen that for catalysts with a higher gas-cracking activity the selectivity shifts towards lighter hydrocarbons. This consecutive cracking prohibits

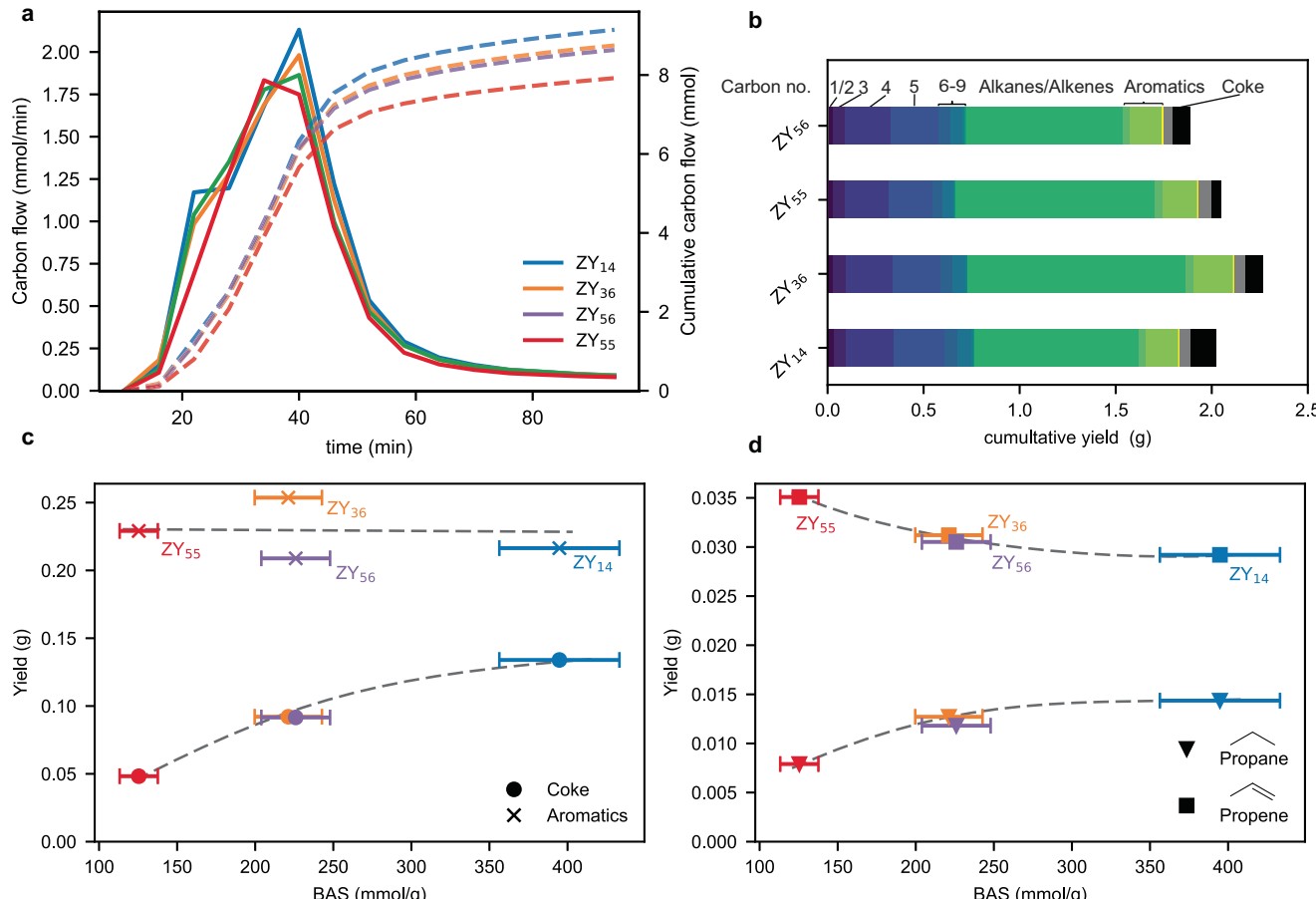

**Fig. 4 | Effect of zeolite acidity on cracking selectivity. a** Evolution of non-condensed hydrocarbons for cracking of polypropylene (PP) using different zeolite Y catalysts in a semi-batch reactor determined by on-line gas chromatography (GC). **b** Cumulative yield of the reaction. Gaseous products were probed by on-line GC, liquid products were characterized by ex-situ GCxGC with flame ionization detection (FID) and mass spectrometry (MS) (see Supplementary Fig. S12 for the corresponding 2D chromatograms), coke deposits were determined by thermogravimetric analysis (TGA). Deviations from 100% yield (2.5 g) are due to minor condensation in cold sections of the autoclave. **c**, **d** Total yield of aromatics, coke, propane, and propene as a function of bulk Brønsted acid site content determined by pyridine infrared (IR) spectroscopy. Error bars taken from the error of the extinction coefficient[55]. Trend lines are drawn to guide the eye.

testing whether preferential cleavage of certain bonds is taking place by means of selectivity analysis in this reactor configuration. We expect that reaction temperature and residence time in the reactor will play a more important role in determining the selectivity pattern. Comparing the selectivity towards propane and propene, a similar observation was made: Propene selectivity is very similar for all catalyst materials, slightly decreasing with increasing bulk BAS content (Fig. 4d). Propene formation also begins early in the reaction (Supplementary Fig. S11), consistent with propene being formed mostly by chain-end scission. The slight decrease of propene formation at increasing bulk acidity might be explained by the increased formation of coke through propene conversion in micropores. By comparison, propane formation occurred later in the process and increased with increasing bulk BAS content, suggesting it is formed in a secondary reaction as well. A more detailed kinetic analysis might be possible with an improved setup in which plastic would be fed at the reaction temperature and on-line detection of higher-boiling hydrocarbons using GCxGC.

In order to develop improved catalyst materials for catalytic cracking of plastics, more specifically polyolefins, valuable structure-composition-performance relationships have to be developed, which are currently mainly lacking in the literature. In this work, we found for a set of zeolite-Y catalysts of different acid site contents a clear mismatch between the performance trends in the cracking of short-chain hydrocarbons and polypropylene. This shows that the structure-

composition-performance relationships for the cracking of short-chain hydrocarbons and polyolefins, such as polypropylene and polyethylene, must be different. A key parameter dictating polyolefin cracking activity is the acid site content outside of the zeolite micropores, which was probed by infrared spectroscopy of substituted pyridine. Furthermore, detailed kinetic analysis relying on isothermal instead of ramped thermogravimetric analysis showed that for highly similar catalyst materials, the polyolefins cracking reaction can exhibit remarkably different sensitivities to catalyst loadings. The quantification of external acid sites, in combination with analysis of catalyst pore structure, activation energies, reaction selectivity, our previous work uncovering accessibility limitations[23], density functional theory calculations, and a simplified simulation of the polyolefins cracking reaction allows us to propose several mechanistic differences between cracking of short-chain hydrocarbons and polymers: The polyolefins cracking reaction is initiated outside of the micropores of the catalyst, and mediated largely by hydride transfer, as opposed to direct alkane activation by the acid sites. Unlike for cracking of short-chain hydrocarbons, the exact location of acid sites on the zeolite external surface becomes important, as a polyolefin molecule can contact multiple acid sites at once. Cracking at different positions along the polymer backbone can lead to large differences in weight-loss kinetics, even though the number of chemical bonds cracked remains the same. Large oligomers of the polyolefin molecule could consecutively enter the zeolite micropores, in which case a combination of high external and

internal acidity could cause a non-linear increase in performance at higher catalyst loadings. Similar effects might also be prevalent for other bulky reactants, e.g., in biomass conversion.

As selectivity towards aromatics and light olefins, such as propene, is very similar for the zeolite-based catalyst materials under study, these products are likely formed directly from the polyolefins, instead of in a secondary reaction from shorter hydrocarbons. The latter, however, is the case for coking. Reducing product residence time in the reactor might, therefore, be critical to reducing catalyst deactivation. To obtain a more detailed picture of the reaction mechanism, further studies will be necessary. Key goals should include probing the reaction kinetics in units of bonds broken, as opposed to weight loss, as well as understanding of the exact distribution of acid sites on the external surface of the zeolite materials. This might be achievable with high-resolution transmission electron tomography of heavy metal ion-exchanged zeolites. Furthermore, probing the selectivity of an isothermal polyolefin cracking reaction using online 2-D chromatography of higher boiling products might enable more mechanistic conclusions.

The results reported imply that besides cracking of lighter hydrocarbons, other activity tests for cracking catalysts, like the micro activity test, which has been correlated to small molecule cracking activity[11], might also fail to reliably relate to polyolefins cracking performance. Therefore, the predictive power of historical hydrocarbon cracking activity data gathered by both academic and industrial researchers will be rather limited. Since a large share of the acid sites are not effectively used, the question can be raised whether zeolite-based materials are the appropriate catalysts for the cracking of polyolefin waste. Utilization of a proper catalyst material with ordered mesopores that could enable higher utilization of the available active sites have been proposed[46], however, the high cost of the surfactants required in their synthesis is most likely prohibitively high[47]. In contrast, zeolites, like zeolite Y, which do not require organic templating agents are comparably cheap: FCC catalysts, which contain around 30 wt% zeolite[3], cost < 3 USD/kg in 2007[48]. The external acidity can be easily increased by steaming, as shown in this work. In combination with the very high bulk acid site content, we, therefore, believe that zeolite-based materials can play a role in the direct cracking of polyolefins, provided their structure and composition can be fully tailored.

## Methods
### Catalyst preparation
The different zeolite materials ($ZY_5$-$ZY_{56}$) were obtained as CBV400, CBV712, CBV720, CBV760, and CBV780 from Zeolyst international. $ZY_5$, $ZY_{36}$, $ZY_{56}$, and $ZY_{55}$ were calcined for 2 h at 550 °C before use. $ZY_{14}$ was transferred to proton form by calcination at 550 °C for 24 h. $ZY_{35st}$ was obtained by mild steaming of $ZY_{14}$ in H-form at 500 °C for 2 h using a saturator at room temperature, and consequently, acid leached using 0.2 M nitric acid at 80 °C for 2 h, and calcined at 550 °C for 2 h.

### Catalyst characterization
**Ar physisorption.** The measurements were conducted at 87 K using a 3 P Sync 400 instrument. All samples were dried at 400 °C under vacuum for 10 h prior to measurement. The surface area was determined by 3 methods: BET (with automated optimization of Rouquerol criteria as implemented in PyGAPS[49]), excess surface work[50], and non-linear density functional theory (NL-DFT). For NL-DFT analysis, a hybrid kernel of spherical micropores and cylindrical mesopores, which for applied to mesoporous ZY previously[51] was fitted to the adsorption branch. All three methods showed the same trends. The total pore volume was determined at $p/p_0$ of 0.99. Micro/mesopore volumes were determined by the t-plot method as implemented in PyGAPS[49] and NL-DFT, as the t-plot method has been shown to underestimate micropore volume for mesoporous zeolites[52]. For the t-plot, a reference isotherm of Ar at 87 K on silica[53] was utilized. 2 nm were used as a cutoff diameter for micropores in NL-DFT. The mesopore volume was calculated as the difference between the total pore volume and the micropore volume. The two methods showed comparable micropore volumes (linear regression showed an $R^2 = 0.96$), with the t-plot method yielding ~ 20% lower volumes, in agreement with prior literature[52].

### Hg porosimetry
The measurements were conducted by 3 P Instruments GmbH using a poremaster 60-GT instrument in a pressure range of 1.5–14.5 MPa. Samples were dried for 12 h at 350 °C under vacuum. A contact angle of 140° was utilized in the analysis.

### Elemental analysis
The silica-alumina ratio was determined by MEDAC Ltd. using inductively coupled plasma-optical emission spectroscopy (ICP-OES).

### $^{27}$Al magic angle spinning nuclear magnetic resonance
Nuclear magnetic resonance (NMR) measurements were performed on an 11.74 T Bruker Avance III spectrometer using a 3.2 mm magic angle spinning (MAS) NMR probe. The MAS NMR spectra were recorded at ambient temperature with a spinning rate of 15 kHz. A resonance frequency of 130.3 MHz was used, and a single-pulse $\pi$/6 excitation with a repetition time of 1 s was applied. The $^{27}$Al chemical shift was externally referenced to an aqueous aluminum nitrate ($Al(NO_3)_3$, Thermo Fisher, +99%) solution. The NMR spectra were processed using a line-broadening of 100 Hz. 2 or 3 Voigt functions were fitted to the data, and the share of framework Aluminum was determined from the integral of the peak centered around 60 ppm. MAS NMR spectra are depicted in Supplementary Fig. S14.

### Scanning electron microscopy
Scanning electron microscopy (SEM) measurements were conducted using a FEI Helios NanoLab G3 UC scanning electron microscope in back-scattered electron mode with a through-the-lens detector. Beam currents of 0.1 nA and 2 kV with dwell times of 5 μs were utilized. A platinum coating was applied with a Cressington 208HR sputter coater to improve conductivity of the samples.

### Acidity characterization
Bulk and external acidity were characterized by infrared (IR) spectroscopy of adsorbed pyridine (Py, Sigma Aldrich, anhydrous, 99.8%) and tri-tert-butyl pyridine (TTBP, Sigma Aldrich, 99%), respectively using a custom setup. Self-supported zeolite pellets (13 mm diameter, pressed at 2 tons) were dried at 500 °C for 1 h under high vacuum ($10^{-5}$ mbar). Pyridine was adsorbed at a pressure of 7–10 mbar for 1 h at 150 °C, with saturation confirmed spectroscopically. Physisorbed species were removed at 150 °C under high vacuum for 30 min. TTBP was adsorbed form the gas phase using a Schlenk flask heated to 80 °C at 0.3 mbar for 30 min, with physisorbed species removed under vacuum at 80 °C. The spectrum of the dry zeolite material at the appropriate temperature was subtracted. The IR spectra were acquired with a Thermo Fischer Nicolet spectrometer at 4 cm$^{-1}$ resolution and averaged over 16 measurements. The measured IR spectra were processed making use of the SpectroChemPy[54] library. The acid site density was quantified according to Eq. 3, where $A$ is the peak area, $S_P$ the surface area of the pellet, $\varepsilon$ the extinction coefficient, and $m_P$ the mass of the pellet.

$$\rho = \frac{A \cdot S_P}{\varepsilon \cdot m_P} \tag{3}$$

For pyridine, and extinction coefficients $\epsilon$ of = 1.54 ± 0.15 cm/μmol and 1.74 ± 0.1 cm/μmol of the ring vibrations located at 1544 cm$^{-1}$ of the protonated pyridine and Lewis acid-coordinated pyridine located at 1455 cm$^{-1}$, respectively[55] were used, for TTBP, the N-H stretch vibration located at 3369 cm$^{-1}$ was quantified, using an extinction coefficient of 5.74 cm/μmol[33]. We note that while the literature extinction coefficient for pyridine was determined gravimetrically[55], it was determined reference to the O-H stretch vibration for TTBP, and we conducted our measurement at higher 80 °C rather than at 25 °C. For temperature-programmed desorption (TPD), pyridine was desorbed at 5 °C/min until 500 °C with continuous recording of the IR spectra. Here, an additional linear baseline correction between 1564 cm$^{-1}$ and 1508 cm$^{-1}$ was applied to the BAS peak.

## Activity testing

**Thermogravimetric analysis experiments.** Polypropylene (PP) of low molecular weight ($M_w$) was obtained from Sigma-Aldrich (Product No. 428116). The reader is advised that the molecular weight of this product varies between batches and is generally higher than labeled. For the TGA experiments, the plastic was first ball milled under N$_2$ atmosphere into a powder to facilitate weight-in. 2 g of PP were milled on a Retsch MM500 vario mixer mill for 1 h at 30 Hz under N$_2$ atmosphere using a 25 ml tungsten carbide jar (Retsch) and 5 ZrO$_2$ grinding spheres (Retsch, 10 mm diameter). The $M_w$ value was determined by size exclusion chromatography (SEC) ($M_w$ = 27,000 g/mol, $M_w/M_n$ = 2.5). The zeolite (pelletized, crushed, and sieved to 150 μm – 212 μm, 0.1–10 mg) was added to a ceramic TGA crucible, and polymer powder (5 ± 0.2 mg) was added on top. The mixture was then heated to 130 °C at 5 °C/min and held for 20 min under N$_2$ atmosphere to remove most of the water adsorbed on the zeolite using a PerkinElmer TGA 8000 instrument. For ramped experiments, the temperature was ramped to 600 °C at 5 °C/min under N$_2$ to crack the plastic, then cooled to 50 °C, and subsequently heated to 800 °C at 20 °C/min to 800 °C to burn off coke deposits. The mass of the catalyst was determined as the weight after the burn-off step, while the mass of the polymer was determined as the difference between the weight after drying and the catalyst weight. For non-isothermal experiments the amount of coke was determined as the difference between weight after cracking and the catalyst weight. Isothermal experiments were conducted analogously, however in the cracking step, the temperature was raised at 200 °C/min to the desired cracking temperature (230–255 °C) and held for 2 h. For the ZY zeolite materials, ≥ 30 isothermal experiments were conducted per catalyst, for the FCC catalyst materials, 9 experiments were performed. The coke amount was not determined as the conversion was not complete in every case. The apparent first-order rate constant was determined by a linear fit as implemented in SciPy[56] to the logarithm of the weight-loss curve in a conversion regime between 25% and 70%.

## Gas cracking experiments

The weight-specific first-order rate constant in cracking of 2,4-dimethylpentane (DMP, TCI, > 99%, < 0.1% olefins verified by GC-FID-MS) was determined in a fixed bed reactor. The catalyst (between 5–20 mg depending on activity, pelletized to 500–612 μm) was dried for 1 h at 550 °C under a He flow. A stream of DMP (> 98% pure, < 0.1% olefins determined by GC-MS-FID) was obtained by using a saturator held at − 7 °C to − 5 °C using N$_2$ as carrier gas and internal standard of a flow of 1.5 mL/min. The stream was then further diluted in a 50 mL/min flow of He. The stream was first sent through a bypass to the online GC to obtain the starting concentration of DMP and subsequently introduced into the reactor at the target reaction temperature of 515 °C. No deactivation was observed over 1 h. The temperature was subsequently lowered stepwise to estimate the activation energy. The rate constant was determined according to Eq. 4, where $m_{cat}$ is the dry weight of the catalyst, $X$ the conversion,

and $\dot{n}_{DMP}$ the molar flow of DMP (Eq. 5, where $\frac{p_{DMP}}{p_0}$ is the partial pressure of DMP determined by GC, $\dot{V}$ is the total volumetric flow, $R$, the gas constant and $T$, the temperature).

$$k_{DMP} = \frac{\dot{n}_{DMP}}{m_{cat}} \ln\left(\frac{1}{1-X}\right) \qquad (4)$$

$$\dot{n}_{DMP} = \frac{p_{DMP}}{p_0} \cdot \frac{\dot{V}}{R \cdot T} \qquad (5)$$

## Semi-batch reactor experiments

The reactor experiments were conducted similarly to a previously published procedure[23,44]. In short, PP and zeolite (2.5 g and 0.675 g, respectively) were loaded into an autoclave reactor and heated under N$_2$ flow at 10 °C/min to 450 °C and held for 1 h using a calibrated power profile. Liquid products were captured using cold traps held at 0 °C, gaseous products were analyzed using a custom-made online gas chromatography-mass spectrometry-flame ionization detector (GC-MS-FID) instrument equipped with sample loops. Liquid products were characterized using an Agilent GCxGC-MS-FID with cryogenic modulation (modulation time of 20 s). The product groups (i.e., alkanes/alkenes, mono/di/tri aromatics, and pyrenes) were assigned by MS. For an estimate of the relative abundance of product groups, the volume of the 2D FID chromatogram was first normalized. A mask based on assigned MS signals was then applied, and the resulting chromatogram was integrated in both dimensions. For details, see the companion Jupiter Notebook. Coke yields were quantified by TGA of the spent catalyst materials (making use of 10 °C/min, and on O$_2$ atmosphere).

## Calculations

**Density functional theory calculations.** The density functional theory (DFT) calculations were performed with the CP2K program set (version 6.1)[57] using the PBE density functional[58] with D3 Grimme corrections[59] and the MOLOPT-DZVP basis set[60] with Goedecker-Teter-Hutter (GTH) pseudopotentials[61]. The basis set and pseudopotentials are distributed with the CP2K program, with the exception of the basis set for La. For this, we used the set made publicly available by Sanliang Ling of the University College London[62].

## Plastic cracking simulation

The Python code for the simulation of weight-loss profiles can be found in the companion Jupyter notebook. 'Polymer molecules' were described by an array of ones and zeros with the former representing carbon atoms and the latter bonds. The starting length of the molecules was set at 200 'carbon atoms', corresponding approximately to the degree of polymerization of a PP with an $M_w$ of 23,000 g/mol. A 'reactor' is represented as an array of molecules. The absolute number of 'molecules' mainly affects the simulation time, in the results shown herein 10 'molecules' were utilized. The steps of the simulation are as follows: First, the probability of cracking a bond is determined by counting the number of bonds in the reactor relative to the starting number of bonds. If a random number is smaller than this probability, a bond will be cracked, otherwise the next timestep is initiated. This enforces first order kinetics in the chemical bonds broken. The molecules in which the bond will be cracked is selected with probability proportional to its size. The bond within the molecule is selected with a probability depending on the selected model (e.g., Gaussian around the middle of the chain). After the 'cracking' of the bond (described by splitting the array in two at the selected bond), 'small molecules' containing 10 carbon atoms or less are removed from the reactor, and the 'weight' is calculated by summing the 'reactor' array. The simulation ran for 400 timesteps and averaged over five repeats.

## Data availability

All raw experimental data generated in this study have been deposited in the OSF database under https://doi.org/10.17605/OSF.IO/PFXH6. Source data are provided in this paper.

## Code availability

A companion Jupyter notebook generating all analyses and figures from raw experimental data can be found on GitHub (Link) and Zenodo[63] and can be executed in Google Colab (Link). We encourage the repurposing of the data reported herein e.g., in more sophisticated kinetic modeling studies.

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

## Acknowledgements

B.M.W. is supported by the Netherlands Organization for Scientific Research (NWO) in the frame of a Gravitation Program, MCEC (Netherlands Center for Multiscale Catalytic Energy Conversion). B.M.W. and I.V. are supported by the Advanced Research Center (ARC) Chemical Buildings Blocks Consortium (CBBC), a public-private research consortium in the Netherlands (https://arc-cbbc.nl). This project was conducted in cooperation with TNO, as part of Brightsite. We thank Rinke Altink (TNO), Mark Roelands (TNO), Florian Meirer (Utrecht University, UU), and Michael Jenks (UU) for helpful discussions. The authors acknowledge Koen Bossers (Sabic) and Nicolaas Friederichs (Sabic) for providing the GPC analysis.

## Author contributions

S.R. conceptualized the specific research in close consultation with I.V., and B.M.W. Z.R. conducted preliminary TGA and gas cracking experiments. Z.B. conducted batch-reactor experiments. J.L. conducted DFT calculations. C.R. conducted Al-NMR measurements. J.M.D conducted SEM measurements. S.R. conducted all remaining experiments, developed the accompanying Jupyter Notebook, and wrote the initial draft. I.V., J.K.vdW., E.T.C.V., and B.M.W. participated in the discussion of the experiments and related results. B.M.W. conceived the overall project and acquired the necessary funding. The manuscript was written through the contribution of all authors. All authors have given approval to the final version of the manuscript.

## Competing interests

The authors declare no competing interests.
